# Role of Diffusion-Weighted Magnetic Resonance Imaging for Characterization of Mediastinal Lymphadenopathy

**DOI:** 10.3390/diagnostics13040706

**Published:** 2023-02-13

**Authors:** Eniyavel Ramamoorthy, Mandeep Garg, Paramjeet Singh, Ashutosh N. Aggarwal, Nalini Gupta

**Affiliations:** 1Department of Radio Diagnosis and Imaging, Post Graduate Institute of Medical Education and Research, Chandigarh 160012, India; 2Department of Pulmonary Medicine, Post Graduate Institute of Medical Education and Research, Chandigarh 160012, India; 3Department of Cytology, Post Graduate Institute of Medical Education and Research, Chandigarh 160012, India

**Keywords:** diffusion weighted imaging, mediastinal lymphadenopathy, metastasis, lymphoma, sarcoidosis, tuberculosis

## Abstract

Background: To assess the diagnostic performance of diffusion-weighted (DW) magnetic resonance imaging (MRI) in the characterization of mediastinal lymph nodes and compare them with morphological parameters. Methods: A total of 43 untreated patients with mediastinal lymphadenopathy underwent DW and T2 weighted MRI followed by pathological examination in the period from January 2015 to June 2016. The presence of diffusion restriction, apparent diffusion coefficient (ADC) value, short axis dimensions (SAD), and T2 heterogeneous signal intensity of the lymph nodes were evaluated using receiver operating characteristic curve (ROC) and forward step-wise multivariate logistic regression analysis. Results: The ADC of malignant lymphadenopathy was significantly lower (0.873 ± 0.109 × 10^−3^ mm^2^/s) than that of benign lymphadenopathy (1.663 ± 0.311 × 10^−3^ mm^2^/s) (*p* = 0.001). When an ADC of 1.0955 × 10^−3^ mm^2^/s was used as a threshold value for differentiating malignant from benign nodes, the best results were obtained with a sensitivity of 94%, a specificity of 96%, and an area under the curve (AUC) of 0.996. A model combining the other three MRI criteria showed less sensitivity (88.9%) and specificity (92%) compared to the ADC-only model. Conclusion: The ADC was the strongest independent predictor of malignancy. The addition of other parameters failed to show any increase in sensitivity and specificity.

## 1. Introduction

Mediastinal lymphadenopathy can be caused by a wide range of diverse pathologies. The differentiation of benign from malignant mediastinal lymphadenopathy poses a formidable challenge, but it is essential for planning treatment and patient management. Computed tomography (CT) is the frequently used imaging modality for the evaluation of various thoracic diseases, such as lung neoplasms and infections [1,2,3], while, lately, magnetic resonance imaging (MRI) has garnered increased attention in thoracic disorders owing to its non-ionizing nature and recent advancements in MR technology [4,5,6].

The characterization of mediastinal lymphadenopathy usually requires invasive sampling; however, CT and conventional MRI have shown variable sensitivity and specificity by relying on morphological criteria, such as the size and enhancement patterns of the lymph nodes [7,8,9,10]. PET (positron emission tomography)-CT is shown to have significantly higher sensitivity (77%) and specificity (86%) in assessing mediastinal metastasis [11,12], but it has a high false-positive rate in reactive and granulomatous diseases [13,14,15]. The lymph-node-specific super paramagnetic contrast agents have shown promising results in the determination of lymph node metastases, but their use is limited due to high costs and a lack of availability [16].

Diffusion-weighted imaging (DWI) is an advanced, functional magnetic resonance imaging (MRI) technique that assesses the Brownian motion of water in living tissues. It is qualitatively represented as diffusion restriction and quantitatively as an apparent diffusion coefficient (ADC) value. The cellular architectural changes in malignant lymph nodes can significantly change the lesion’s diffusion characteristics and ADC value. It is an excellent non-invasive method to distinguish between benign and malignant mediastinal lymphadenopathy [17]. This will immensely aid in patient care, as it will not only avoid many unwanted invasive tests but also be useful in starting early treatment. A few studies have already demonstrated the use of DWI in the characterization of mediastinal lymphadenopathy [18,19,20,21]. In this study, we included morphological criteria of the lymph nodes along with an ADC value to evaluate the performance of diffusion-weighted and conventional MRI parameters.

## 2. Materials and Methods

### 2.1. Patient Enrollment

This prospective study was approved by the institutional review board (INT/IEC/2015/27). A total of 43 treatment-naive patients with mediastinal lymphadenopathy as detected on chest CT were included in this study between January 2015 and June 2016. Informed, written consent was taken from all subjects. The patients who refused to give consent, those having contraindications for MRI (claustrophobic, non-compatible medical implants, etc.), where lymph nodes were not seen on DW-MRI, and in whom tissue diagnosis could not be obtained were excluded from the study. The remaining patients underwent DW-MRI and T2 Half-Fourier Acquisition Single-Shot Turbo Spin Echo (HASTE). A flow diagram of the study population is shown in Figure 1.

### 2.2. MR Imaging

MRI was performed within 15 days of the CT scan on a 1.5 T MRI scanner (Magnetom Aera system, Siemens Healthcare, Erlangen, Germany) equipped with a body coil and a high-performance gradient system. Axial HASTE images were obtained (repetition time (TR) msec/effective echo time (TE) msec, 500/38; section thickness, 4 mm; voxel size, 1 × 1 × 4 mm^3^; field of view 333 mm; band width, 781 hz/px, average time of acquisition 2 min). All DW-MRI was performed using a single-shot echo-planar imaging sequence (TR/TE msec 5100/56; section thickness, 6 mm; b values, 0, 400 and 1000 s/mm^2^; voxel size, 1.3 × 1.3 × 6 mm^3^; SPAIR fat suppression; field of view 400 mm; average time of acquisition 2 min 50 s). Respiratory triggering was not utilized. Breath holds of 20–25 s were used to acquire T2 HASTE images, and DWI was acquired during free breathing.

### 2.3. Lymph Node Sampling and Pathologic Examinations

Endo-bronchial ultrasound (EBUS)-guided fine needle aspiration cytology (FNAC) (n = 26) and trans-bronchial biopsy (n = 12) were carried out for the para-tracheal and sub-carinal lymph nodes. CT-guided FNAC was carried out (n = 5) if the lymph nodes were seen in the anterior mediastinum. Two patients required repeat EBUS-FNAC and one repeat transbronchial biopsy, which gave the final results in our study. No patient underwent a mediastinoscopy or any other invasive procedure to reach the diagnosis. The locations of these lymph nodes were labeled according to the International Association for the Study of Lung Cancer (IASLC) classification [22,23]. In each patient, the largest lymph node was sampled and matched with the MRI parameters.

### 2.4. Image Analysis

The MRI-DWI dataset was transferred to a workstation (Siemens Leonardo) and systematically evaluated by one experienced radiologist having experience of more than 15 years in thoracic radiology and another trainee radiologist in his final year of a 3-year period of radiology residency. Both observers were blinded to the histopathological diagnosis and other details of the cases. Both observers evaluated the largest lymph nodes in all patients.

### 2.5. Quantitative Analysis

The short-axis diameter (SAD) of the largest lymph node was recorded. If conglomerated lymph nodes were encountered, the largest mass was evaluated for the purpose of analysis. The ADC value of the largest lymph node was recorded by drawing the region of interest (ROI) placed on the largest lymph node, excluding calcification. The area of the ROI circle was measured with a range of 0.5–2.5 cm^2^. Though the software gives the value of three ADC values, namely minimum, mean, and maximum, we only used the mean value in this study. Three ROIs were drawn in the same lymph node in different axial sections and the mean of three ADC values was calculated and represented in ×10^−3^ mm^2^/s.

### 2.6. Qualitative Analysis

Necrosis was defined as a T2 hyper-intense signal comparable to cerebrospinal fluid in the dorsal spine. The T2 signal intensity (SI) of the largest mediastinal lymph node was recorded as heterogeneously hyper-intense by the presence of necrosis. Lymph nodes without necrosis were coined as homogenously hyper-intense. The signal intensities of enlarged lymph nodes on DWI (at b = 400, 1000) and ADC maps were recorded as the presence or absence of diffusion restriction by both observers.

### 2.7. Statistical Analysis

The Statistical Packages for the Social Sciences (SPSS software version 20, Chicago, IL, USA) was used for the statistical analysis. The quantitative indices SAD and ADC were compared using a Student’s *t*-test. A Chi-square test was applied to look for an association between the pathology of the lymph nodes with diffusion restriction and T2 heterogeneity. A receiver operating characteristic (ROC) curve was used to determine the cut-off points of ADC value and SAD of the lymph nodes to discriminate malignant lymph nodes from benign. Cronbach’s alpha was calculated to assess the internal consistency between both observers in the measurement of ADC values and SAD. The inter-operator consistency between the assessment of the diffusion restriction of mediastinal lymph nodes and the T2 heterogeneity of the lymph nodes was evaluated using the kappa statistic. The added value of ADC was compared with other MRI parameters using multivariate logistic regression analysis.

## 3. Results

The mean age of participants was 49 years (age range, 26–80 years). The study comprised 17 cases of malignant lymphadenopathy and 26 patients with benign lymph nodes (Table 1). Cronbach’s alpha value and kappa statistic showed good internal consistency in the evaluation of lymph node characters.

### 3.1. SAD of Lymph Nodes

The mean lymph nodal SAD was 2.45 cm, with a range of 0.8 to 6.2 cm. The smallest lymph node (measuring 8 mm) was identified as reactive in cytology. The SAD of malignant and benign lymph nodes showed a significant difference by Student’s *t*-test (*p* < 0.001) with a poor sensitivity of 66.7% and a specificity of 92%, with an area under the curve (AUC) of 0.820.

### 3.2. T2Heterogeneity

A homogeneous T2 hyper-intense signal was seen in twenty-six lymph nodes (60.5%; five malignant, and twenty-one benign), and a heterogeneous hyper-intense signal was seen in seventeen lymph nodes (39.5%; twelve malignant, and five benign). Lymph nodes showing T2 heterogeneity were seen in 4/9 (44.4%) patients of tuberculosis, while only 1/12 (8.3%) patients of sarcoidosis showed T2 heterogeneity. T2 heterogeneity was more common in the malignant lymph nodes (*p* = 0.001). T2 heterogeneity showed a sensitivity of 72.2%, a specificity of 84%, and a positive predictive value of 76.5%, and a negative predictive value of 80.8%.

### 3.3. Diffusion Restriction

Out of forty-three lymph nodes, ten benign lymph nodes showed no diffusion restriction (two—TB, four—sarcoidosis, four—reactive). Thirty-three lymph nodes showed diffusion restrictions (seventeen malignant, sixteen benign). All the malignant lymph nodes showed diffusion restriction irrespective of pathology with 100% sensitivity and negative predictive value. However, it showed poor specificity (40%) and positive predictive values (54.5%).

### 3.4. ADC

The mean ADC values of all lymph nodes were 1.332 × 10^−3^ mm^2^/s, with the lowest ADC value in lymphoma (0.791 × 10^−3^ mm^2^/s) and the highest ADC value in tuberculosis (2.085 × 10^–3^ mm^2^/s). The mean ADC values for different etiologies of the lymph nodes are shown in Table 2.

The representative T2-weighted images, DWI, and ADC maps of the malignant and benign lymph nodes are shown in Figure 2 and Figure 3 respectively.

There was a significant difference between the ADC of malignant lymph nodes (0.873 ± 0.109 × 10^−3^ mm^2^/s) and benign lymph nodes (1.663 ± 0.311 × 10^−3^ mm^2^/s) (*p* = 0.001). The difference in distribution of ADC values of benign and malignant lymph nodes is depicted in a box and whisker plot (Figure 4). ROC curve analysis also revealed that an ADC cut-off value of 1.095 × 10^−3^ mm^2^/s could differentiate between benign and malignant lymph nodes with 94% sensitivity, 96% specificity, and an AUC of 0.996 (Figure 5).

### 3.5. Multivariate Logistic Regression Analysis

A model only containing ADC had a similar sensitivity (94.6%) and specificity (96%) as a model containing all four parameters with *p* < 0.001. A comparative model containing other MRI criteria, namely T2 heterogeneity, a SAD value of the largest lymph node, and the presence of diffusion restriction, showed less sensitivity and specificity (Table 3).

## 4. Discussion

The differentiation between benign and malignant mediastinal lymphadenopathy on imaging has always been a challenge. DW-MRI has emerged as a promising non-invasive tool in this direction. The absolute ADC value on DW-MRI has shown higher specificity and sensitivity than any other parameters, such as the SAD of lymph nodes, the T2 heterogeneous signal, and diffusion restriction. In a study by de Bondt RB et al., the T2 heterogeneous signal intensity of cervical lymph nodes showed significantly better diagnostic accuracy compared to the other morphological parameters [24]. We also found that T2 heterogeneity was significantly more common in malignant disease in our subset of patients, but it showed less specificity compared to ADC values.

Donners R et al. reported ADC values of normal intra-thoracic lymph nodes to be ~1.53 × 10^−3^ mm^2^/s [25]. They also observed that the visualization of normal intra-thoracic lymph nodes was relatively uncommon in whole-body DW imaging of healthy individuals. This could be the reason for the non-visualization of lymph nodes in four of our patients, whom we had to exclude from the study.

The role of DW imaging in the characterization of mediastinal lymphadenopathy has been evaluated in multiple studies [18,19,20,21]. These studies showed that the ADC value of malignant lymph nodes was significantly different from the benign lymph nodes. Similar evaluations were also conducted in mediastinal masses [26] and pediatric populations [27]. Abou Youssef H et al. also reported significant differences in the mean ADC values of lymphoma and sarcoidosis. However, they did not evaluate other morphological criteria of lymph nodes and their comparison with ADC values [28].

In accordance with our study, Abdel Razek et al. also observed significant differences not only in the ADC values of benign and mediastinal lymph nodes but also in diffusion tensor parameters, such as mean diffusivity and fractional anisotropy [29].

Mean ADC values of benign (1.663 × 10^−3^ mm^2^/s) and malignant (0.873 × 10^−3^ mm^2^/s) lymph nodes in our study were different from a previous study by Abdel Razek et al. [18]. This is probably due to differences in the imaging protocol, as the b value of our study protocol (b = 0, 400, 1000 s/mm^2^) was different from the one used by Abdel Razek et al. (b = 0, 300, 600 s/mm^2^).

In our cohort, tuberculous lymph nodes were shown to have a higher mean ADC value (1.614 × 10^−3^ mm^2^/s), which is also different from the results reported by Naranje P et al. (1.29 × 10^−3^ mm^2^/s) [30]. This is again probably due to the difference in the b values used in the two studies, as we included b values of 0, 400, and 1000 s/mm^2^, while b values of 0, 400, and 800 s/mm^2^ wereused by Naranje P et al. AbouKhadrah RS et al. observed that the high b values of 800 and 1000 s/mm^2^ were of higher significance than the b values of 0, 50, and 400 s/mm^2^ in differentiating benign from malignant masses in the head and neck regions [31].

The ADC value of lymphoma was slightly lower than other metastatic lymph nodes in our study, but it was not statistically significant. Furthermore, no significant difference was seen in ADC values of lymph nodes of Hodgkin and non-Hodgkin lymphoma in the current study, while Sabri YY et al. reported significant differences in the ADC value of Hodgkin and non-Hodgkin lymphoma patients in a previous study conducted with 32 subjects [32]. These observations perhaps need further validation with more studies with a larger sample size.

Using ROC analysis, we were able to distinguish between malignant and benign lymph nodes based on ADC values. Our ADC cut-off value (1.0955 × 10^−3^ mm^2^/s) was comparable to the one reported by de Bondt RB et al. (1.0 × 10^−3^ mm^2^/s) and high b values were used in both of these studies (b = 1000), even though the anatomical regions of the evaluation were different [33].

In multivariate logistic regression analysis, we compared the ADC with the rest of the parameters by the forward stepwise method. We found that the model containing all other variables, excluding ADC, showed less sensitivity and specificity. Adding morphological variables with ADC failed to show any increase in sensitivity and specificity. Though the other criteria, such as T2 heterogeneity, showed good sensitivity and specificity, these were inferior to the ADC. Our results were similar to the results obtained in the study conducted by de Bondt RB et al. in cervical lymphadenopathy [33].

The meta-analyses conducted by Peerlings J et al. and Shen G et al. showed the high diagnostic performance of ADC, but they expressed a lack of consensus regarding differences in imaging protocols and parameters for performing DWI [34,35]. We used a 1.5-tesla scanner with echo-planar imaging, similar to most of the studies in these meta-analyses.

DWI can provide a paradigm shift in lung cancer care by helping in the sub-classification of nodal staging in lung cancer. This gains more significance in light of the recent re-proposal of recommendations given by the IASLC Staging Committee for considering the sub-categories of the N-descriptor (based on both anatomic and quantitative criteria) [36]. The results of a recent study conducted by Bertoglio P et al. also partially validated that the sub-classification of N-disease can improve the stratification of patients [37]; thus, DWI can play an important role in selecting the correct treatment plan.

DWI could also be used as an imaging biomarker for cancer surveillance. It can identify early quantitative metric changes due to treatment response in cancer patients. Positive treatment responses for many tumor types can be detected by assessing an early increase in ADC values and low pre-treatment ADC values [38].

Our study has some limitations: First, the sample size of the study was small. Additional studies with a larger sample size are thus needed to understand the exact diagnostic accuracy of DWI and ADC values. Second, in our study cohort, most of the assessed lymph nodes were larger than one centimeter. Hence, we could not evaluate the MR characteristics of sub-centimetric lymph nodes.

## 5. Conclusions

The ADC value was the single most important parameter in differentiating between benign and malignant lymph nodes in the mediastinum. The addition of other morphological parameters, such as lymph node size and T2 heterogeneity, did not improve the diagnostic performance of MRI. However, further prospective and multicentric studies are needed to validate the diagnostic accuracy of DWI, along with the standardization and optimization of MRI sequences.

## Figures and Tables

**Figure 1 diagnostics-13-00706-f001:**
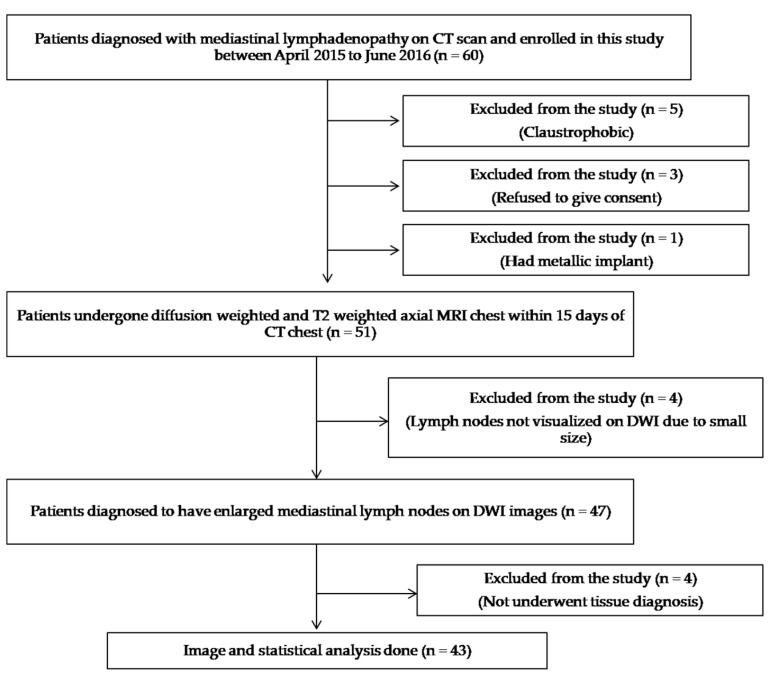
Flow diagram depicting the recruitment process of the study cohort. [CT—Computed tomography, DWI—diffusion-weighted imaging].

**Figure 2 diagnostics-13-00706-f002:**
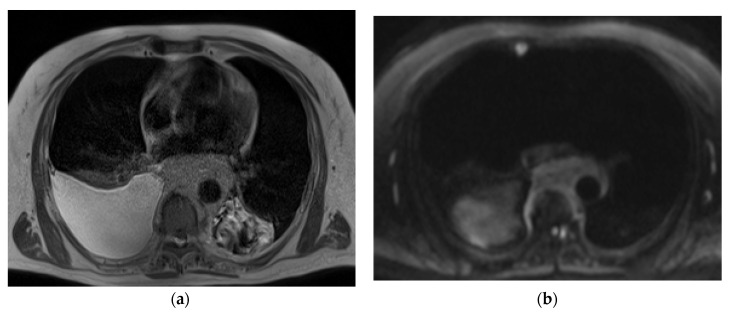
(**a**) T2-weighted axial MRI of a 75-year-old male shows a T2 homogeneous signal in enlarged lymph nodes encasing descending thoracic aorta; (**b**) Diffusion-weighted axial MRI shows diffusion restriction in most of the lymph node; (**c**) System generated ADC map of corresponding lymph node shows ADC mean value of 0.829 × 10^−3^ mm^2^/s. The final pathological diagnosis was lymphoma.

**Figure 3 diagnostics-13-00706-f003:**
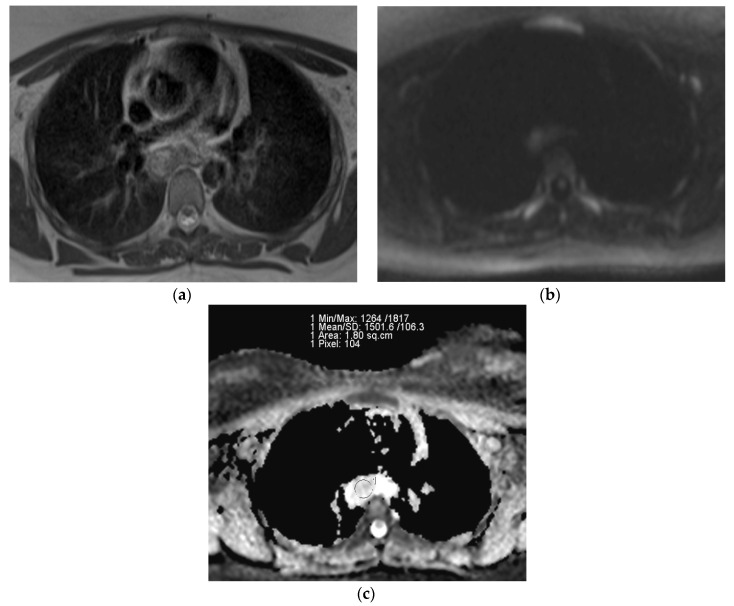
(**a**) T2-weighted axial MRI of 26-year-old female shows T2 heterogeneity in lymph nodes at subcarinal location (station 7); (**b**) Diffusion-weighted axial MRI shows mild central diffusion restriction; (**c**) Systemgenerated ADC map of corresponding lymph node shows ADC mean value of 1.501 × 10^−3^ mm^2^/s. The final pathological diagnosis was tuberculosis.

**Figure 4 diagnostics-13-00706-f004:**
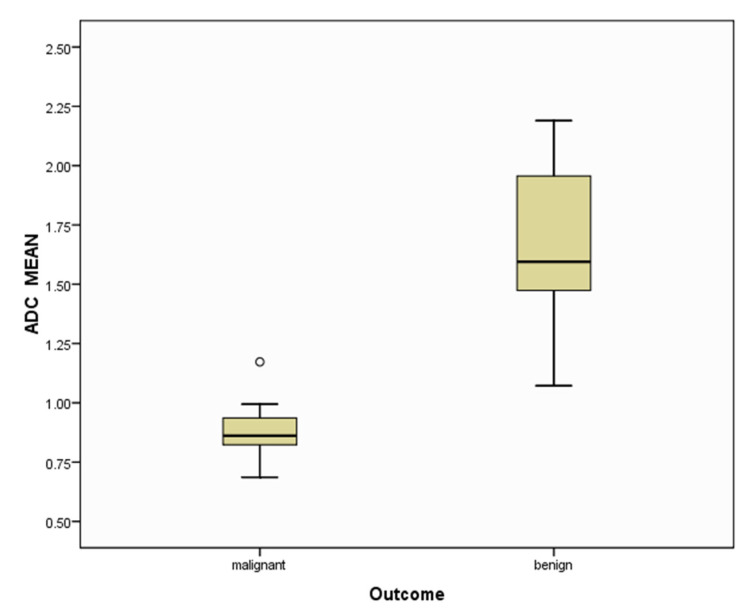
Box and whisker plot of the ADC mean value of the benign versus malignant mediastinal lymph nodes.°-denotes the maximum ADC mean value in the malignant group.

**Figure 5 diagnostics-13-00706-f005:**
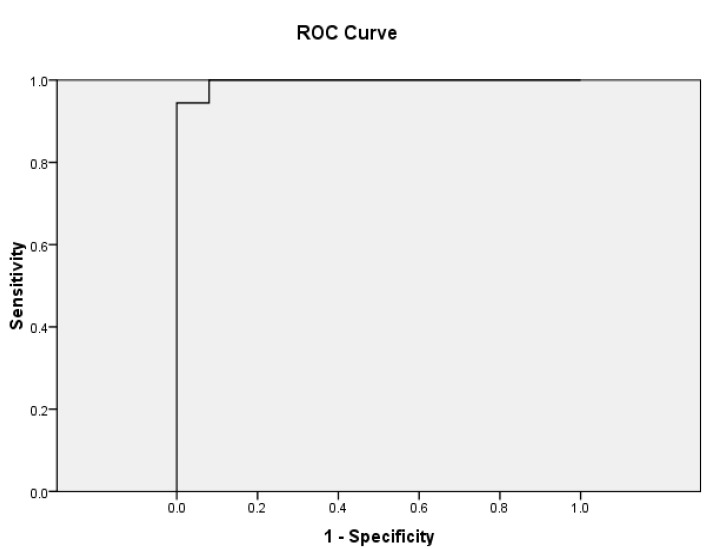
ROC curve of ADC values to differentiate malignant and benign lymphadenopathy. (AUC = 0.996; SE = 0.006; asymptotic significance < 0.001; 95% confidence interval 0–1.0). ROC; receiver operating characteristic, AUC; area under the curve, SE; standard error.

**Table 1 diagnostics-13-00706-t001:** Histological characteristics of evaluated lymph nodes.

Histological Findings	Value
Malignant lymph nodes	17 (39.5%)
Metastasis (small cell lung cancer)	2 (4.6%)
Metastasis (non-small cell lung cancer)	7 (16.2%)
Metastasis (oropharyngeal cancer)	1 (2.3%)
Metastasis (unknown primary)	1 (2.3%)
Non-Hodgkin lymphoma	4 (9.3%)
Hodgkin lymphoma	2 (4.6%)
Benign lymph nodes	26 (60.5%)
Sarcoidosis	12 (27.9%)
Tuberculosis	9 (20.9%)
Reactive lymph nodes	5 (11.6%)

Note—Data are numbers of lymph nodes in particular histopathology (total number = 43), with percentages in parentheses.

**Table 2 diagnostics-13-00706-t002:** ADC mean values of lymph nodes from different etiologies.

Lymph Nodal Histopathology	Mean * ADC Value (×10^−3^ mm^2^/s)
Malignant lymph nodes	0.873 ± 0.109
Metastasis	0.9155
Lymphoma	0.8283
Benign lymph nodes	1.663 ± 0.311
Sarcoidosis	1.6085
Tuberculosis	1.6140
Reactive lymph nodes	1.925

* ADC—apparent diffusion coefficient.

**Table 3 diagnostics-13-00706-t003:** Sensitivity and specificity of different parameters for the diagnosis of malignant lymphadenopathy for the models containing: ADC mean (≤1.095 × 10^−3^ mm^2^/s versus >1.095 × 10^−3^ mm^2^/s), SAD (short axis diameter), T2 heterogeneity, and diffusion restriction.

Model	SENSITIVITY (%)	SPECIFICITY (%)
ADC mean only	94	96
ADC mean with all other criteria	94	96
T2 heterogeneity and SAD	83	92
T2 heterogeneity, SAD, and diffusion restriction	88	92

ADC—apparent diffusion coefficient; SAD—short axis diameter.

## Data Availability

Data are available from the authors.

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
