# Peer review of "Role of Diffusion-Weighted Magnetic Resonance Imaging for Characterization of Mediastinal Lymphadenopathy"

_diagnostics, 2023, doi:10.3390/diagnostics13040706_

Round 1

Reviewer 1 Report

The article is well written but I have few suggestions

1. The latest references of last five years are very few in fact one in number. Few more latest articles can be added .

2. There is no ref.  mentioned in the text at line 209 where there is reference of Donners et al. 

3. Text alignment is not proper in Table 1. 

Author Response

Response to Reviewer 1

Point 1: The latest references of last five years are very few in fact one in number. Few more latest articles can be added.

Response: As per reviewer’s suggestions, latest references have been added as following:

  1. Raafat TA, Ahmed SM, Seif EMA, Mikhael HSW, Awad AS. Role of diffusion-weighted MRI in characterization of mediastinal masses. Egyptian Journal of Radiology and Nuclear Medicine. 2020 Sep 21;51(1).
  2. Razek AAKA, Gaballa G, Elashry R, Elkhamary S. Diffusion-weighted MR imaging of mediastinal lymphadenopathy in children. Japanese Journal of Radiology. 2015 Jun 12;33(8):449–54
  3. Abou Youssef H, Elzorkany M, Hussein S, Taymour T, Gawad M. Evaluation of Mediastinal Lymphadenopathy by Diffusion Weighted MRI; Correlation with Histopathological Results. Advances in Respiratory Medicine. 2019 Jun 28;87(3):175–83
  4. Abdel Razek AAK, Baky KA, Helmy E. Diffusion Tensor Imaging in Characterization of Mediastinal Lymphadenopathy. AcadRadiol. 2022 Feb;29 Suppl2:S165-S172. 
  5. Naranje P, Singh R, Bhalla A, Pandey S. Magnetic resonance imaging in response assessment of mediastinal tuberculous lymphadenopathy: Going beyond size. Lung India2021;38:431.
  6. Aboukhadrah RS, Imam HH. Multiple b values of diffusion-weighted magnetic resonance imaging in evaluation of solid head and neck masses. Egyptian Journal of Radiology and Nuclear Medicine. 2019 Nov 26;50(1)
  7. Sabri YY, Ewis NM, Zawam HEH, Khairy MA. Role of diffusion MRI in diagnosis of mediastinal lymphoma: initial assessment and response to therapy. Egyptian Journal of Radiology and Nuclear Medicine. 2021 Aug 31;52(1).
  8. Bertoglio P, Ricciardi S, Alì G, Aprile V, Korasidis S, Palmiero G, et al. N2 lung cancer is not all the same: an analysis of different prognostic groups†. Interactive CardioVascular and Thoracic Surgery. 2018 May 19;27(5):720–6.

Point 2: There is no ref.  mentioned in the text at line 209 where there is reference of Donners et al. 

Response: We thank the reviewer for highlighting this omission. We have now added the reference number as suggested.

Donners R et al. [19] reported ADC values of normal intra thoracic lymph node to be 1.53 × 10− 3mm2/s. They also observed that visualisation of normal intra thoracic lymph nodes was relatively uncommon in whole-body DW imaging of healthy individuals. This might be the reason for the non-visualization of lymph nodes in our 4 patients, whom we had to exclude from the study.

Point 3: Text alignment is not proper in Table 1. 

Response: Text alignment in Table 1 has been corrected in the revised manuscript.

Reviewer 2 Report

This manuscript investigates the diagnostic value of DTI metrics for mediastinal lymphadenopathy. The paper is generally well written. However, it has been shown that the lymph node size (short axis diameter) gives as suboptimal criterion to indicate metastases (Toloza et al 2003). In my opinion the current manuscript doesn’t provide additional scientific advances compared to previous research (e.g. Razek et al. 2022 and earlier work).

Additionally, I have a few points to make:

1.       Line 45-46: ‘It is represented qualitatively as diffusion restriction and quantitatively as an apparent diffusion coefficient value.’: ADC is one of the quantitative metrics that can be extracted from diffusion weighted MRI. Is diffusion restriction here mentioned the same as the commonly referred mean diffusivity? Please explain.

2.       Line 79: How many directions were used for each b-value? Could fractional anisotropy be also evaluated?

3.       Line 103-104: The authors report an interval, not an average.

4.       Line 129 These are not results, they are part of the methodology

5.       T2 heterogeneity (lines 141-148): Needs a figure to compare homogeneous vs heterogeneous T2 hyperintense signal.

6.       Subsections ADC (lines 161-179) and ROC (lines 180-198) are a repeat of observations in previous results subsections. They should be integrated.

7.       Lines 225-226: High b-values to increase the specificity and negative predictive value: Unclear, please elaborate.

8.       Line 243: planar instead of planner

References:

Toloza, E.M., Harpole, L. and McCrory, D.C., 2003. Noninvasive staging of non-small cell lung cancer: a review of the current evidence. Chest123(1), pp.137S-146S.

Razek, A.A.K.A., Baky, K.A. and Helmy, E., 2022. Diffusion Tensor Imaging in Characterization of Mediastinal Lymphadenopathy. Academic Radiology29, pp.S165-S172.

Author Response

Response to Reviewer 2

Point 1: This manuscript investigates the diagnostic value of DTI metrics for mediastinal lymphadenopathy. The paper is generally well written. However, it has been shown that the lymph node size (short axis diameter) gives as suboptimal criterion to indicate metastases (Toloza et al 2003). In my opinion the current manuscript doesn’t provide additional scientific advances compared to previous research (e.g. Razek et al. 2022 and earlier work).

Response 1: We thank the reviewer for his comments. Our manuscript has investigatedthe diagnostic  vaule of diffusion weighted magentic resonance imaging. We did not evlaute DTI metrics in our study. We additionally compared  the diagnostic value of ADC  with other morphological criterias in our study. In our study, we concluded the superiority of ADC value over other morphological criteria like lymphnodal size and T2 hetergeneity.

Point 2:Line 45-46: ‘It is represented qualitatively as diffusion restriction and quantitatively as an apparent diffusion coefficient value.’: ADC is one of the quantitative metrics that can be extracted from diffusion weighted MRI. Is diffusion restriction here mentioned the same as the commonly referred mean diffusivity? Please explain.

Response 2: Yes, we have referred diffusion restriction as reduced diffusivity in our manuscript.

Point 3:Line 79: How many directions were used for each b-value? Could fractional anisotropy be also evaluated?

Response 3: As our study did not include the DTI parameters, so fractional anisotropy etc. was not evaluated.

Point 4:Line 103-104: The authors report an interval, not an average.

Response 4: The area of the ROI circle to measure the ADC value was between 0.5–2.5 cm2 and corrected as interval in present submission. The software calculated three ADC values and we used the mean value in this study. This information is there in the ‘material and method section’ under ‘quantitative analysis’.

Point 5:Line 129These are not results, they are part of the methodology

Response 5: We agree with the reviewer’s comment and the line containing demographic data of subjects from ‘results’ section has been removed.

Point 6:T2 heterogeneity (lines 141-148): Needs a figure to compare homogeneous vs heterogeneous T2 hyperintense signal.

Response 6: We thank the reviewer for his comments. Homogenous and heterogeneous T2 hyperintense signal in lymph nodes has been shown inFigure 2a and Figure 3a respectively.

Point 7:Subsections ADC (lines 161-179) and ROC (lines 180-198) are a repeat of observations in previous results subsections. They should be integrated.

Response 7: We thank the reviewer for his observation. And as suggested, we have integrated these lines in the revised text as:

There was a significant difference between the ADC of malignant lymph nodes (0.873±0.109×10-3 mm2/s) and benign lymph nodes (1.663±0.311×10-3 mm2/s) (P=0.001).The significant difference in distribution of ADC values of benign and malignant lymph nodes depicted in Box and whisker plot (Figure 4). ROC curve analysis also revealed that an ADC cut off value of 1.095×10-3 mm2/s could differentiate between benign and malignant lymph nodes with 94% sensitivity, 96% specificity, and an AUC of 0.996 (Figure 5)”.

Point 8:Lines 225-226: High b-values to increase the specificity and negative predictive value: Unclear, please elaborate.

Response 8: We thank the reviewer for highlighting this point and we have reframed this statement in the revised text. The use of high b-value is considered better in differentiating benign from malignant lesions. Aboukhadrah RS et al. observed that the high b-value of 800 and 1000 sec/mm2 was of higher significance than the b-value of 0, 50 and 400 sec/mm2 in differentiation benign from malignant masses in head and neck region. We have also added this reference in the study [reference number 31].

Point 9:Line 243: planar instead of planner

Response 9:planner was corrected to planar. 

Reviewer 3 Report

Dear Authors and Editors

 I had the opportunity to review this interesting manuscript entitled “Role of diffusion-weighted magnetic resonance imaging for characterization of mediastinal lymphadenopathy”.

This study highlighted the role of ADC as independent predictor of malignancy compared to other parameters as non-invasive method to distinguish between benign and malignant mediastinal lymphadenopathy.

This article is very well-written but there are some concerns and issues that should be evaluated.

1)      In the cohort of patients enrolled there are to many pathologies, both malignant and benign, that usually show different behavior regarding the nodal involvement. Moreover, it is not clear how many patients had an oncological anamnesis and how many patients had a lung cancer at the time of the MRI. More tables are needed to clarify this aspect.

2)      A comparison of MRI result with the PET/CT results should be added to better evaluate e compare the MRI role.

3)      TBNA and EBUS are, in most cases, insufficient in the diagnosis of Lymphoma or benign lesion. It could be interesting report how many patients underwent to more invasive procedure with diagnostic aim.

4)      In discussion authors should report more in details the potential of MRI also in the oncological surveillance.

5)      How many patients with lung cancer underwent surgery?. Have the authors compared the pathological examination of the lymph nodes with MRI findings.

Authors could improve the discussion by adding a paragraph on the role on MRI in lung cancer patients (All patient of NSCLC need an accurate evaluation of the mediastinum) as well as on the pattern of nodal involvement (N1-2 or 3). I suggest to improve discussion with the following article: PMID: 29788107

Author Response

Response to Reviewer 3

Point 1: In the cohort of patients enrolled there are to many pathologies, both malignant and benign, that usually show different behavior regarding the nodal involvement. Moreover, it is not clear how many patients had an oncological anamnesis and how many patients had a lung cancer at the time of the MRI. More tables are needed to clarify this aspect.

Response 1:We thank the reviewer for this observation.

Our study cohort comprised 17 cases of malignant lymphadenopathy [metastasis from small cell (n=2) and non-small cell lung cancer (n=7), metastasis from oropharyngeal cancer (n=1) and unknown primary (n=1), non-Hodgkin (n=4) and Hodgkin lymphoma (n=2),]. In this malignant group, only 9 paitents were having lung cancers at the time of MRI.

Similarly, there were 26 patients with benign lymph nodes [tuberculosis (n=9), sarcoidosis (n=12), and reactive lymph nodes (n=5)].

We have added a new table (Table 1) to elaborate these details.

Point 2:A comparison of MRI result with the PET/CT results should be added to better evaluate compare the MRI role.

Response 2: We agree with the reviewer’s comment. But PET/CT was not done in our study cohort.

Point 3:TBNA and EBUS are, in most cases, insufficient in the diagnosis of Lymphoma or benign lesion. It could be interesting report how many patients underwent to more invasive procedure with diagnostic aim.

Response 3:We agree with the reviewer’s comments. Inour study cohort, EBUS was done in n = 26, TBNA in n = 12 and CT guided FNAC in n =5.Only 2 patients required repeat FNAC and one trans-bronchial biopsy which gave final results in our study. No patient underwent mediastinoscopy, thoracotomy or any other invasive procedure to reach at the diagnosis.

This has been included in the revised text in ‘Material and Method’ section under ‘lymph node sampling and pathologic examinations’

Point 4:In discussion authors should report more in details the potential of MRI also in the oncological surveillance.

Response 4:As suggested by the reviewer, we have added following one paragraph in discussion –

Diffusion-weighted imaging could also be used as an imaging biomarker for cancer surveillance. It can identify early quantitative metric changes due to treatment response in cancer patients. Positive treatment responses for many tumor types can be detected by assessing an early increase in the apparent diffusion coefficient (ADC) values and low pre-treatment ADC values [30].

Point 5:How many patients with lung cancer underwent surgery? Have the authors compared the pathological examination of the lymph nodes with MRI findings.

Response 5:We thank the author for his comment. To the best of our knowledge, only 2 patients underwent surgery in a private center outside our institute, and hence, we couldn’t compare the pathological examination of lymph nodes with MRI. Rest of the patients got palliative treatment only.

Point 6:Authors could improve the discussion by adding a paragraph on the role on MRI in lung cancer patients (All patient of NSCLC need an accurate evaluation of the mediastinum) as well as on the pattern of nodal involvement (N1-2 or 3). I suggest to improve discussion with the following article: PMID: 29788107

Response 6:We agree with the reviewer’s comments and have also cited the suggested article. We have added following new paragraph in discussion:

DWI can provide a paradigm shift in lung cancer care by helping in sub-classification of nodal staging in lung cancer. This gains more significance in the light of recent re-proposal of recommendations given earlier by IASLC Staging Committee for considering the sub-categories of N-descriptor (based on both anatomic and quantitative criteria) [36]. The results of a recent study done by Bertoglio P et al. have also partially validated that sub-classification of N-disease can improve stratification of patients [37]; thus DWI can play an important role in selecting the correct treatment plan.

Round 2

Reviewer 3 Report

Dear Authors and Editors, 

 I really appreciated the effort made to improve the quality of the manuscript.

Authors have properly answered to all my concerns so I have no more question or issue to address.

Best regards

Best regards